# Overcoming TRAIL-resistance by sensitizing prostate cancer 3D spheroids with taxanes

**Korie A. Grayson[1,2], Nidhi Jyotsana[2], Nerymar Ortiz-Otero[1,2], Michael R. King[2]\***

**1** Meinig School of Biomedical Engineering, Cornell University, Ithaca, New York, United States of America, **2** Department of Biomedical Engineering, Vanderbilt University, Nashville, Tennessee, United States of America

\* mike.king@vanderbilt.edu

**Data Availability Statement:** Data from this paper have been archived with Data Archiving and Networked Services (DANS), and is available at the following link: https://doi.org/10.17026/dans-289-3jxr.

## Abstract

Three-dimensional spheroid cultures have been shown to better physiologically mimic the cell-cell and cell-matrix interactions that occur in solid tumors more than traditional 2D cell cultures. One challenge in spheroid production is forming and maintaining spheroids of uniform size. Here, we developed uniform, high-throughput, multicellular spheroids that self-assemble using microwell plates. DU145 and PC3 cells were cultured as 2D monolayers and 3D spheroids to compare sensitization of TRAIL-resistance cancer cells to TRAIL mediated apoptosis via chemotherapy based on dimensionality. Monocultured monolayers and spheroids were treated with soluble TRAIL alone (24 hr), DTX or CBZ alone (24 hr), or a combination of taxane and TRAIL (24 + 24 hr) to determine the effectiveness of taxanes as TRAIL sensitizers. Upon treatment with soluble TRAIL or taxanes solely, monolayer cells and spheroids exhibited no significant reduction in cell viability compared to the control, indicating that both cell lines are resistant to TRAIL and taxane alone in 2D and 3D. Pretreatment with CBZ or DTX followed by TRAIL synergistically amplified apoptosis in 2D and 3D DU145 cell cultures. PC3 spheroids were more resistant to the combination therapy, displaying a more additive effect in the DTX + TRAIL group compared to 2D. There was a downregulation of DR4/5 expression in spheroid form compared to monolayers in each cell line. Additionally, normal fibroblasts (NFs) and cancer-associated fibroblasts (CAFs) were cocultured with both PCa cell lines as spheroids to determine if CAFs confer additional resistance to chemotherapy. We determined that co-cultured spheroids show similar drug resistance to monocultured spheroids when treated with taxane plus TRAIL treatment. Collectively, these findings suggest how the third dimension and cocultures of different cell types effect the sensitization of androgen-independent prostate cancer cells to TRAIL, suggesting therapeutic targets that could overcome TRAIL-resistance in metastatic castration-resistant prostate cancer (mCRPC).

## Introduction

Over the last decade, 3D cell culture has become more appealing due to the physiologically relevant interactions and environmental cues that closely mimic the complex dimensionality *in*

**Funding:** MRK was awarded Grant No. R01CA203991 from the National Institutes of Health, National Cancer Institute: nih.gov The sponsors had no role in the study design, data collection and analysis, decision to publish, or preparation of the manuscript.

**Competing interests:** The authors have declared that no competing interests exist.

*vivo* [1, 2]. This increased dimensionality is lost in traditional 2D cell culture, which is a mainstay in research. Although 2D cell culture is inexpensive and highly reproducible, the method does not take into account the cell-cell and cell-matrix interactions along with communication mechanisms attributed to 3D structures [1, 2]. In studying cancer biology, 3D cell culture recapitulates the tumor cell and spatial heterogeneity seen in solid tumors along with tumor proliferation, responsiveness, and metastasis [3, 4]. From proliferating and quiescent cells to cancer cells and cancer stem cells, tumor spheroids offer a proficient way to recreate solid tumors *in vitro*. 2D cell cultures are unable to reproduce the nutrient, oxygen, and pH gradients seen in solid tumors that effect responses to anti-cancer therapeutics [4]. 3D cell culture methodologies can be implemented in high-throughput screenings (HTS) of therapeutic candidates to enhance drug development at pre-clinical stages and minimize the use of genetically different animal models [5, 6]. There are now numerous ways to make 3D spheroids that are scaffold-based and scaffold-free [2]. We have previously used a scaffold-free method of polydimethylsiloxane (PDMS) cured multi-well plates to make 3D tumor spheroids [7]. Our lab has shown that 3D breast cancer spheroids have increased binding to E-selectin and exhibit more migratory and invasive characteristics when in a 3D state [8, 9].

There is a shift from using conventional and unspecific anti-cancer therapies towards treatment strategies that selectively target cancer cells. Tumor necrosis factor (TNF)-related apoptosis-inducing ligand (TRAIL) is a molecule that preferentially induces apoptosis in cancer cells without affecting normal healthy cells. Apoptosis is induced when TRAIL binds to TRAIL-R1 (DR4) and TRAIL-R2 (DR5) on the surface of the cell [10]. The death receptors trimerize and activate the extrinsic receptor-mediated apoptotic pathway by recruiting Fas-associated death domain (FADD), which in turn activates caspase-8 to form a death-inducing signaling complex (DISC). Caspase-8 activates two pathways, intrinsically through the mitochondria employing cytochrome c, and a more direct route extrinsically which signals directly to the executioner caspases 3 [11]. Unfortunately, several tumor cell lines have been shown to be resistant to TRAIL or develop mechanisms to block TRAIL-induced apoptosis. We have previously explored TRAIL-resistance in 3D spheroids and found breast cancer cells cultured as tumor spheroids are more resistant to TRAIL-mediated apoptosis through the downregulation of DR4/5 [12]. 3D breast cancer spheroids also contained a subpopulation of breast cancer stem cells that lacked the DR4 expression needed to initiate the apoptotic initiation indicating that the heterotypical environment of spheroids can profoundly affect sensitivity to TRAIL-mediated apoptosis.

In separate work, we are exploring a sensitization mechanism of prostate cancer (PCa) cells to TRAIL via pretreatment with taxanes for it has been previously shown that some PCa cell lines exhibit resistance to TRAIL alone [13]. Taxanes are microtubule stabilizers that inhibit disassembly of microtubules to render cells in mitotic arrest and further prevent further cell division and growth [14, 15]. Docetaxel (DTX) and cabazitaxel (CBZ) are the two chemotherapies approved to treat metastatic castration-resistant prostate cancer (mCRPC) [14, 16]. These chemotherapies have helped extend patient survival by several months; however, chemoresistance inevitably ensues and patients are left with limited treatment options leading to tumor progression, recurrence, and metastasis [17, 18].

We have moved to exploring resistance mechanisms of PCa cells in multicellular 3D environments. Tumor microenvironments are heterogenous and consist of different cell types such as endothelial cells, stromal cells, and immune cells. Studies have shown that stromal cells in the tumor microenvironment (TME) help promote PCa progression and spread [19, 20]. One of the main components of the TME are fibroblasts, specifically cancer-associated fibroblasts (CAFs). CAFs are differentiated fibroblasts, derived from the stromal compartment of prostate tumors, that overexpress fibroblast activation protein (FAP), α-smooth muscle actin

(SMA), and fibroblast specific protein 1 (FSP-1) [21, 22]. CAFs are recognized as playing a critical role in PCa progression by prompting tumor proliferation, therapy resistance, epithelial-to-mesenchymal transition (EMT), and invasiveness [23, 24].

Recently, our group showed that newly-activated CAFs induce shear resistance to prostate tumor cells via intercellular contact and soluble derived factors [25]. In this current study, we hypothesized that 3D cocultured PCa spheroids would exhibit more resistance to TRAIL-induced apoptosis. We investigated the effect of a third dimension in spheroid responsiveness to taxane plus TRAIL synergism. We confirmed that pretreatment with taxanes significantly enhances the susceptibility of DU145 spheroids more than PC3 spheroids to TRAIL-induced apoptosis compared to taxane or TRAIL alone despite the incorporation of CAFs. This preferential synergistic effect based on cell line can be mediated by DR4 and DR5 expression. Here, we show that therapeutic response changes from 2D to 3D microenvironments and fibroblast incorporation to show similar resistance to taxane plus TRAIL therapy.

## Materials and methods

### Chemicals/Reagents

Cabazitaxel (ADV465749196) and docetaxel (01885) were purchased from Sigma Aldrich (St. Louis, MO, USA). Both taxanes were dissolved in a 1:10 solution of DMSO:PBS (pH 7.4) to a final concentration of 100 μM. Dimethyl sulfoxide (DMSO) was purchased from ATCC. Soluble histidine-tagged TNF-related apoptosis-inducing ligand (TRAIL) (BML-SE721-0100) was purchased from Enzo Life Sciences (Farmingdale, NY, USA).

### Cell lines and culture conditions

Androgen-independent prostate cancer cell lines DU145 (ATCC #HTB-81) and PC3 (ATCC #CRL-1435), androgen-dependent prostate cancer cell line LNCaP (ATCC #CRL-1740), prostate-derived fibroblasts WPMY-1 (ATCC #CRL-2854) and hTERT PF179T (ATCC #CRL-3290) were purchased from American Type Culture Collection (Manassas, VA, USA). DU145 cells were maintained in Eagle's Minimum Essential Medium (EMEM) cell culture media (Corning, Corning, NY, USA). PC3 cells were cultured in F12 cell culture media (Gibco, ThermoFisher, Waltham, MA, USA). LNCaP cells were cultured in RPMI-1640 culture media (ATCC 30–2001). WPMY-1 normal fibroblast cells were cultured in Dulbecco's Modified Eagle's Medium (DMEM). Media was supplemented with 10% (v/v) fetal bovine serum (FBS) and 1% (v/v) Pen Strep (both purchased from Gibco) under humidified conditions at 37°C and 5% $CO_2$. hTERT PF179T cancer-associated fibroblast cells were maintained in EMEM supplemented with 10% (v/v) FBS, 1% (v/v) sodium bicarbonate 7.5% solution (#SH30033.01, GE Life Sciences Hyclone), and 0.01% 10 mg/mL puromycin (#A11138-03, Gibco).

### Spheroid production

Cells were harvested from culture flasks using 0.25% trypsin (Gibco). Cancer cells were stained with either CellTracker Deep Red dye (#C34565, ThermoFisher) or CellTracker Blue CMAC dye (#C2110, ThermoFisher), and fibroblasts were stained with CellTracker Green CMFDA dye (#C2925, ThermoFisher) in respective serum free media using a working concentration of 25 μg/mL for 30 min at 37°C. Cells were washed twice with DPBS and then seeded as monocultures or mixed with fibroblasts for cocultures and then seeded into AggreWell™800 micro-well plates (#34815) following the manufacturer's protocol (STEMCELL Technologies, Cambridge, MA, USA). Monoculture spheroids were seeded at 1.5x10^6 cells per well. Cocultured spheroids were seeded at 1.5x10^6 cells per well and at a 2:1 ratio of cancer cells to

fibroblasts. Methocult™ H4100 (#04100) was added to media in each well to reach a 0.25% methylcellulose concentration (STEMCELL). AggreWell™ plates were spun down at 100 x g for 3 minutes to capture cells in the microwells. Plates were incubated at 37˚C and 5% $CO_2$ for 48 hr until spheroid formation was complete.

## Cancer stem cell identification in surviving fraction

Cancer stem cells (CSCs) were identified as CD44+/CD24- subpopulations. Following the apoptosis assay protocol described previously, cells from each treatment group were also labeled using anti-CD44-APC (338805) and anti-CD24-Brilliant Violet 421™ (311121) monoclonal antibodies (BioLegend San Diego, CA, USA). Mouse IgG1 APC (400119) and mouse IgG2a Brilliant Violet 421™ (400259) constituted isotype controls (BioLegend San Diego, CA, USA). The following control samples were used to calibrate the instrument: unlabeled, Annexin-V only, PI only, AV/PI, mouse IgG1 APC only, mouse IgG2a Brilliant Violet 421™ only, anti-CD44-APC only, and anti-CD24-Brilant Violet 421™. Flow cytometry plots were analyzed using FlowJo software (FlowJo, Ashland, OR, USA).

## Small interfering RNA (siRNA) transfection

siRNA oligonucleotide sequences were used to target DR4 (M-008090-02-0005 siGENOME Human TNFRSF10A (8797) siRNA—SMARTpool) and DR5 (M-004448-00-0005 siGENOME Human TNFRSF10B (8795) siRNA—SMARTpool). A negative control siRNA (Scr; D-001210-01–05 siGENOME NonTargeting siRNA #1) was used to control for siRNA delivery effects. SiRNA reagents were purchased from Horizon Discovery (Lafayette, CO, USA). Cells were plated in 12-well plates and transfected using the siRNAs, Lipofectamine RNAIMAX reagent (Invitrogen), and Opti-MEM (Gibco) according to the manufacturer's instructions. Cell medium was replaced after 48 hours and then cells were treated with taxane (0.25 μM CBZ or DTX, 24 hr), and TRAIL (100 ng/mL, 24 hr) either alone or in combination (24+24 hr = 48 hr). The final concentration for each siRNA was 30 nM. To determine extent of knockdown via flow, PE-anti-human DR4 (307206) and PE-anti-human DR5 (307406) were used (BioLegend, San Diego, CA, USA).

## Apoptosis assay

Monocultured and cocultured spheroids were plated into AggreWell™ plates for 48 hr and then exposed to taxane alone (CBZ or DTX at 1μM, 24 hr), TRAIL alone (400 ng/mL, 24 hr), or taxane plus TRAIL for 24+24 hr = 48 hr. Treatment concentrations were scaled up the difference in media volume between 2D and 3D cell culture wells. Cancer cells were stained with CellTracker Deep Red or Blue, and fibroblasts were stained with CellTracker Green. Cells were harvested using Accutase™ (STEMCELL Technologies) and analyzed with Annexin V/PI flow cytometry assay to assess cell viability. FITC Annexin V (556419) and propidium iodide (556463) staining solutions were purchased from BD Biosciences (San Diego, CA, USA). Staining of untreated and treated cells was carried out according to the manufacturer's instructions. Cells were incubated for 15 min with Annexin V reagents at RT in the dark and immediately analyzed using a Guava EasyCyte 12HT benchtop flow cytometer (Millipore Sigma). Viable cells were classified as AV-/PI-, early apoptotic cells as AV+/PI-, late-stage apoptotic cells as AV+/PI+, and necrotic cells as AV-/PI+. Flow cytometry plots were analyzed using FlowJo software (FlowJo, Ashland, OR, USA). The following control samples were used to calibrate the instrument: unlabeled cell samples to evaluate the level of autofluorescence and adjust the instrument accordingly, cells labeled with Cell Tracker, and cell samples labeled individually with Annexin-V and PI to define the boundaries of each cell population.

## AlamarBlue assay

Ten thousand cells were seeded in 100 µl of media in each well of a 96-well flat-bottom transparent plate. One-tenth of the volume of AlamarBlue reagent (G-Biosciences, 786–921) was directly added to the wells and incubated for 4 hr at 37˚C in a cell culture incubator, shielded from direct light. Results were recorded by measuring fluorescence using a fluorescence excitation wavelength with a peak excitation of 570 nm and a peak emission of 585 nm on a microplate reader (Tecan Infinite F500, Tecan Group Ltd.).

## Cell cycle analysis

An equal number of cells was trypsinized and fixed in cold 70% ethanol for 10 min and then stained with propidium iodide (PI) solution (1 µg/µL PI and 0.125% RNaseA; Sigma Aldrich, St. Louis, MO) at room temperature for 15 min. 10,000 cells per sample were analyzed using Guava easyCyte flow cytometer (Millipore, Billerica, MA, USA).

## Synergistic evaluation by Jin's formula

The synergistic effect of combined taxane and TRAIL was analyzed using Jin's formula [26, 27]. The formula is $Q = E_{a+b}/(E_a + E_b - E_a \times E_b)$, where $E_{a+b}$, $E_a$ and $E_b$ are the average inhibitory effects of the combination treatment, taxane only and TRAIL only, respectively. In this method, $Q < 0.85$ indicates antagonism, $0.85 < Q < 1.15$ indicates additive effects and $Q > 1.15$ indicates synergism. The $E_{a+b}$, $E_a$ and $E_b$ quantities were obtained from the apoptosis assay.

## Western blotting

Monolayers and spheroid cultures were treated accordingly: DMSO (vehicle control), CBZ, or DTX for 24 hr. Afterwards, cells were rinsed with sterile PBS and lysed with 4x Laemmli sample buffer (Bio-Rad #1610747) and then subjected to sodium dodecyl sulfate-polyacrylamide gel electrophoresis (SDS-PAGE) [7% (w/v) for DR4 and DR5] and transferred to PVDF membranes. After transfer, membranes were blocked with 5% milk (Boston BioProducts, Ashland, MA, USA) in tris-buffered saline supplied with 0.1% Tween (Thermo Fisher Scientific). Primary antibodies were prepared at 1:500 dilution in 5% milk in the case of DR5 (Abcam ab199357). In the case of DR4 (Abcam ab8414) and GAPDH (Millipore MAB374) primary antibody was prepared at 1:5000 dilution in 5% milk. Anti- rabbit secondary antibody conjugated to horseradish peroxidase (Rockland, Pottstown, PA, USA) was prepared at 1:2000 dilution in 5% milk. Membranes were imaged with West Pico (Thermo Fisher Scientific) per their respective protocols, using an ImageQuant LAS-4000 system (GE Healthcare, Chicago, IL, USA).

## Confocal microscopy for spheroid characterization

Monolayers, monocultured spheroids, and co-cultured spheroids were imaged in bright field mode using an Olympus IX81 motorized inverted microscope. Image J software was used to evaluate the number and diameter of spheroids formed. Confocal images were taken of monocultured and cocultured spheroids using an LSM 710 Meta inverted confocal microscope to determine composition. Tumor and fibroblast cells were enumerated using the following criteria: cancer cells (CellTracker Deep Red) and fibroblasts (CellTracker Green) positive for nuclear staining with DAPI (blue).

## Statistical analysis

GraphPad Prism 8 (San Diego, CA, USA) software was used to plot and analyze data sets. Two-tailed unpaired t-test was used for comparisons between two groups, with $p < 0.05$ considered significant. ANOVA was used for comparing multiple groups with $p < 0.05$ considered significant. Data are presented as mean ± SD with at least three independent replicates used for each experiment.

# Results

## Role of DR4 and DR5 receptors in influencing sensitivity to taxane plus TRAIL treatment in PCa 2D monolayer cells

A dose-response curve of varying concentrations (0–2μM) of each taxane for 24 hr and treatment with 100 ng/mL of TRAIL for 24 hr was generated to characterize the degree of apoptosis in DU145 and PC3 cells (Fig 1A). From the dose-response curve, we identified the most effective concentration to be 0.25 μM to produce a significant response of TRAIL-induced apoptosis. Pretreating cells with taxanes for 24 hr followed by TRAIL significantly increased apoptosis from ~20% when treated with TRAIL alone to ~45% when treated with 0.0078 μM + TRAIL and finally to ~80% when treated with 0.25 μM ₊ TRAIL (Fig 1A). Further increasing the concentration did not show an additional effect on cell viability for the sequential therapy.

To determine cell viability when exposed to taxanes alone, cells were treated with each taxane at 0.25 μM over a 24–96 hr period to assess apoptosis over prolonged exposure times (Fig 1B). The cells exhibited a time-dependent response to each taxane, demonstrating decreasing cell viability over time, with DU145 cells showing increased cell death compared to PC3 cells. From this time-response curve, we identified the most effective time (24 hr) for treatment to sensitize both PCa cells to TRAIL-induced apoptosis, rather than taxane-induced apoptosis. Taxanes can inhibit cell growth arrest, therefore cell proliferation was explored and measured using a propidium iodide-based cell cycle assay to capture growth inhibition. A clear shift towards the left was observed in the histograms in cells treated with 0.25 μM CBZ or DTX, indicating cell growth arrest (Fig 1C and 1D).

Previous studies and ongoing work in our lab have examined changes in death receptor expression in response to taxane therapy [13, 32]. To establish the role of DR4 and DR5 receptors in influencing sensitivity to taxane + TRAIL response in PCa cells, DU145 and PC3 cells were transfected with siRNA targeting DR4 and DR5 to knockdown receptor expression. The efficiency of siRNA knockdown in each cell line was confirmed via flow cytometry. There was a clear knockdown in DR5 expression observed following siRNA treatment; however, DR4 knockdown was not observed possibly due to DR4 expression being less pronounced in the cell lines (Fig 1E and 1F). DU145 and PC3 cells were treated with TRAIL alone, taxane alone, or taxane + TRAIL to calculate the degree of TRAIL sensitization after DR4 and DR5 knockdown. DR5 knockdown inhibited the combination treatment in DU145 and PC3 cells as detected by the AlamarBlue cell viability assay that evaluates cell proliferation (Fig 1G and 1H). In terms of apoptosis via Annexin V/PI assay, DR5 knockdown inhibited the combination treatment in DU145 cells and not PC3 cells (Fig 1I and 1J).

Experiments were also conducted in one androgen receptor-positive prostate caner cell line, LNCaP, for comparison. Cell proliferation was measured using a propidium iodide-based cell cycle assay to capture growth inhibition. A clear shift towards the left was observed in the histograms of cells treated with 0.25 μM CBZ or DTX, indicating cell growth arrest (Fig 2A). The efficiency of siRNA knockdown in LNCaP was confirmed via flow cytometry. There was a clear knockdown in DR5 expression following siRNA treatment; however, DR4 knockdown

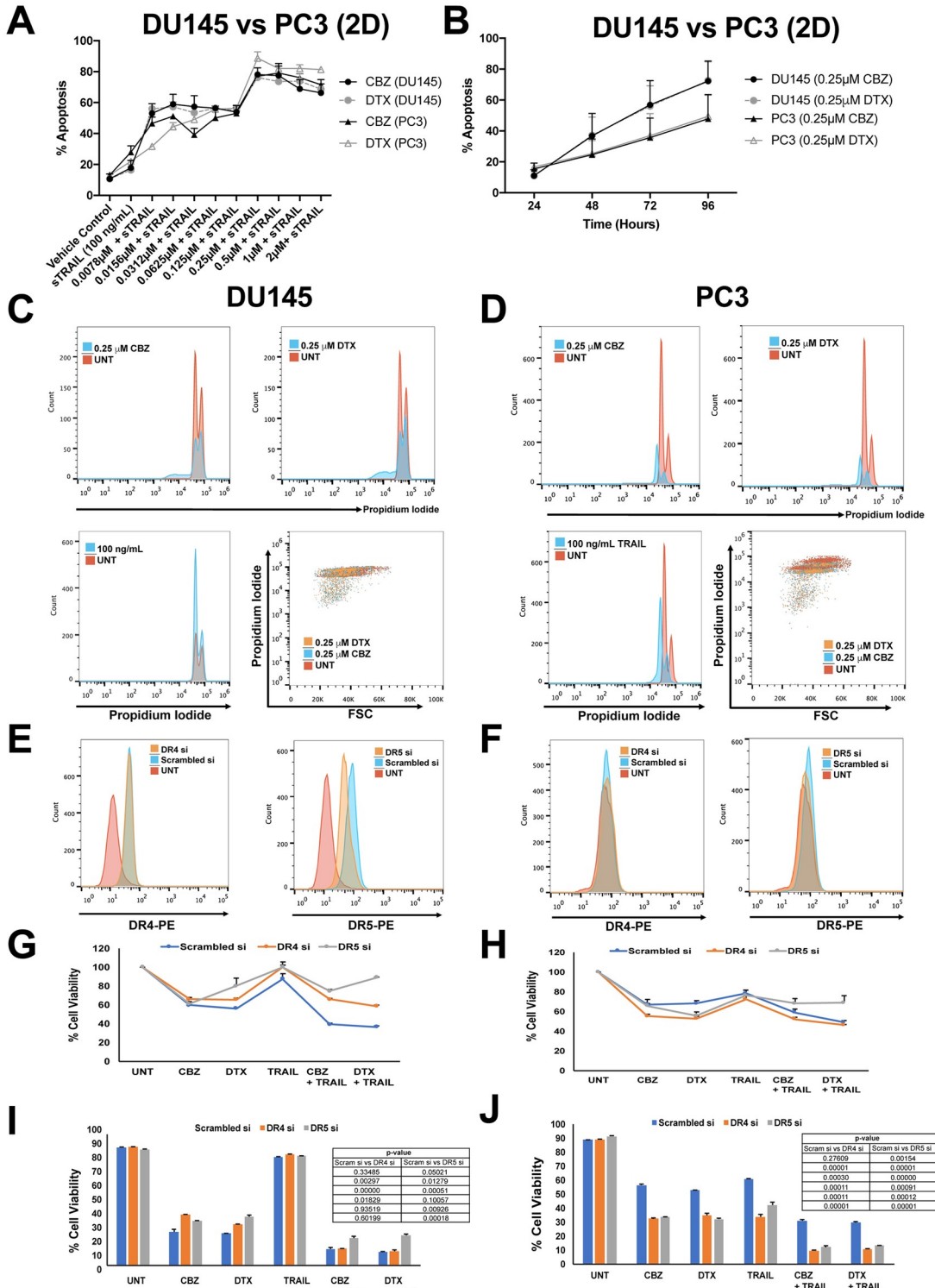

**Fig 1. Synergistic effect of taxane + TRAIL treatment in DU145 and PC3 cells with DR4 and DR5 knockdown in 2D cell culture. A**. Percentage of apoptosis observed in DU145 and PC3 cells when cells were treated with varying concentrations of taxane plus 100 ng/mL TRAIL. **B**. Percentage of apoptosis observed after taxane treatment of DU145 and PC3 cells at 24, 48, 72, and 96 hr after initiation of treatment. **C, D**. Representative propidium iodide histograms of DU145 and PC3 cells treated with either 0.25 μM CBZ, 0.25 μM DTX, or 100 ng/mL TRAIL. **E, F**. Representative histograms of death receptor expression of DR4 and DR5 siRNA knockdown compared to scrambled and untreated controls. **G, H**. Cell viability of DU145 and PC3 cells after

DR4 and DR5 knockdown when treated with 0.25 µM CBZ, 0.25 µM DTX, 100 ng/mL TRAIL, CBZ + TRAIL, or DTX + TRAIL and assessed via AlamarBlue assay. **I, J**. Cell viability of DU145 and PC3 cells after DR4 and DR5 knockdown when treated with 0.25 µM CBZ, 0.25 µM DTX, 100 ng/mL TRAIL, CBZ + TRAIL, or DTX + TRAIL and assessed via Annexin V/PI apoptosis assay. The values represent the mean ± SD (n = 3).

was not observed possibly due to DR4 expression being less pronounced in this cell line as well (Fig 2B). LNCaP cells were treated with TRAIL alone, taxane alone, or taxane + TRAIL to calculate the degree of TRAIL sensitization after DR4 and DR5 knockdown. DR5 knockdown inhibited the combination treatment in LNCaP cells as detected by the AlamarBlue cell viability assay that evaluates cell proliferation (Fig 2C). In terms of apoptosis via Annexin V/PI assay, DR5 knockdown did not inhibit the combination treatment in LNCaP cells (Fig 2D). Overall LNCAP cells seemed more sensitive to all treatment groups but most noticeably TRAIL. Due to this observation, we did not proceed with making LNCaP spheroids due to their inherent susceptibility rather than the resistance to TRAIL compared to DU145 and PC3 cell lines.

## Characterizing mono- and cocultured PCa spheroids with fibroblasts

In order to optimize and characterize spheroid formation, hormone-insensitive DU145 and PC3 prostate cancer cells were cocultured in AggreWell™800 plates with CAFs (PCa:CAF) and NFs (PCa:NF). Mono- and cocultured spheroids were plated in the microwell plate using centrifugation to force aggregation of a defined number of cells to control spheroid size and uniformity. Cocultured spheroids were seeded at $1.5 \times 10^6$ cells per well and at a 2:1 ratio of cancer

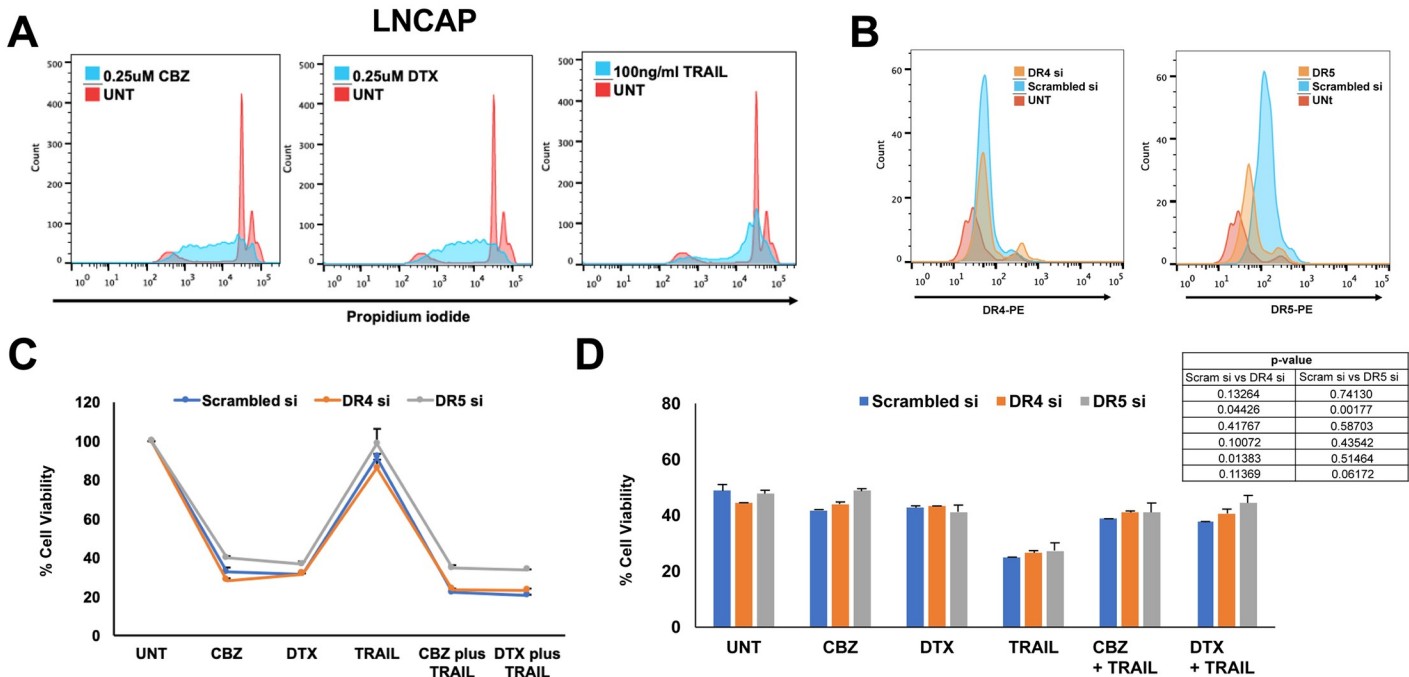

**Fig 2. Synergistic effect of taxane + TRAIL treatment in LNCaP cells with DR4 and DR5 knockdown in 2D cell culture. A**. Representative propidium iodide histograms of LNCaP cells treated with either 0.25 µM CBZ, 0.25 µM DTX, or 100 ng/mL TRAIL. **B**. Representative histograms of death receptor expression of DR4 and DR5 siRNA knockdown compared to scrambled and untreated controls. **C**. Cell viability of LNCaP cells after DR4 and DR5 knockdown when treated with 0.25 µM CBZ, 0.25 µM DTX, 100 ng/mL TRAIL, CBZ + TRAIL, or DTX + TRAIL and assessed via AlamarBlue assay. **D**. Cell viability of LNCaP cells after DR4 and DR5 knockdown when treated with 0.25 µM CBZ, 0.25 µM DTX, 100 ng/mL TRAIL, CBZ + TRAIL, or DTX + TRAIL and assessed via Annexin V/PI apoptosis assay. The values represent the mean ± SD (n = 3).

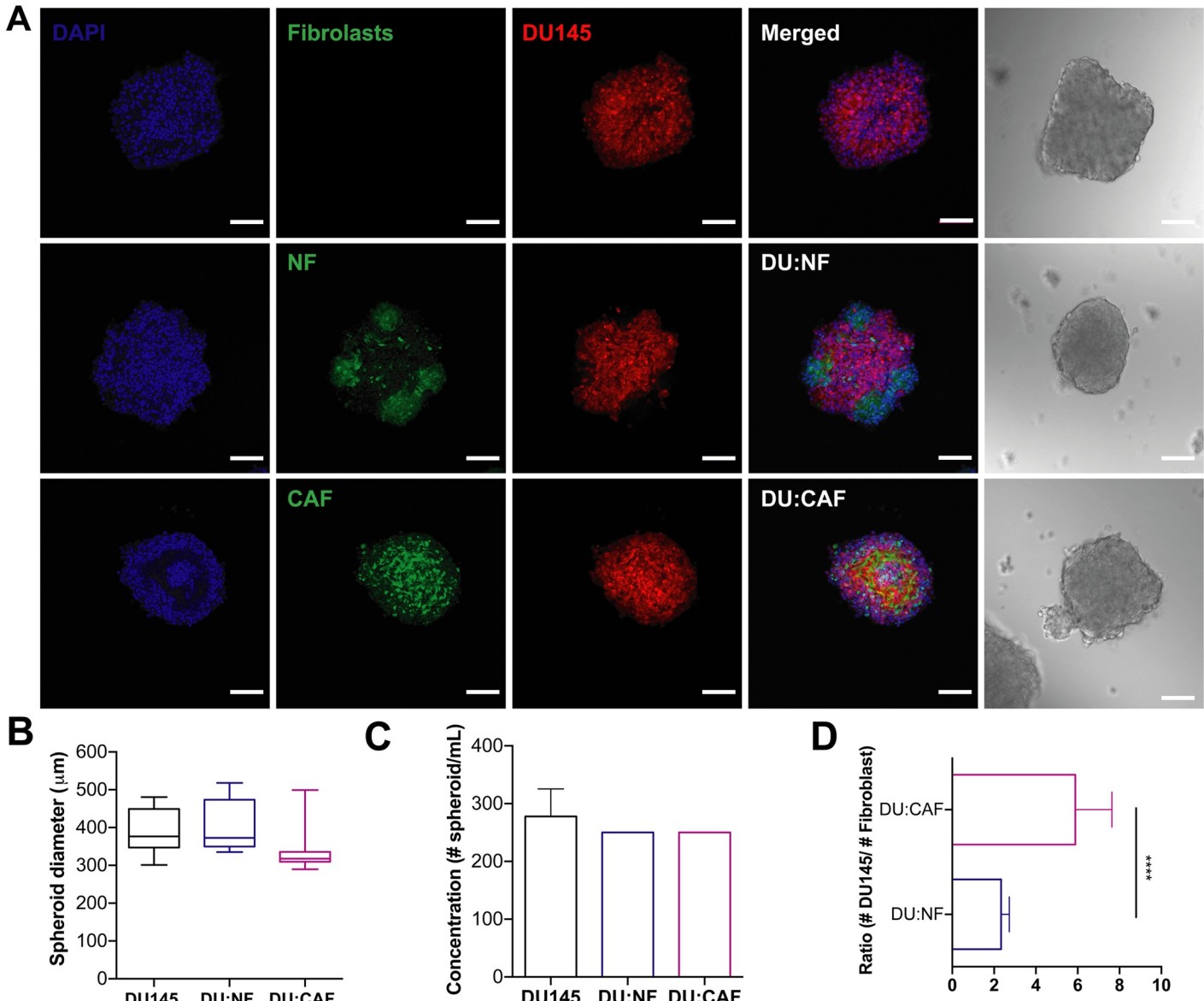

**Fig 3. DU145 spheroids monocultured and co-cultured with prostate fibroblasts in AggreWell™800 plates yielded consistent and reproducible spheroids. A.** Brightfield image and confocal images of DU145 spheroids after 48 hr incubation. Scale bar = 100 µm. **B.** Diameters of DU145, DU145-NF, and DU145-CAF spheroids. **C.** Concentration of spheroids/mL in DU145, DU145-NF, and DU145-CAF cultures. **D.** Ratio of DU145 cells to fibroblasts in cocultured spheroids. The values represent the mean ± SD (n = 3), **** p < 0.0001.

cells to fibroblasts due to the larger size of fibroblasts compared to cancer cells. We confirmed successful spheroid formation with confocal microscopy and brightfield images (Fig 3A). DU145 monocultured spheroids formed compact spheroids with an average diameter size of 390 µm. DU:NF cocultured spheroids formed compact spheroids with an average diameter size of 401.3 µm.

We observed NFs aggregating at certain sites around the spheroid rather than complete incorporation into a spheroid structure. DU:CAF cocultured spheroids also formed compact spheroids with an average diameter size of 338 µm. No significant difference was observed between the sizes of DU145, DU:NF, and DU:CAF spheroids (Fig 3B). These DU145, DU:NF,

and DU:CAF spheroids showed an average concentration of 277, 250, and 250 spheroids/mL, which does not exceed the hypothetical yield of 300 spheroids for 1 well of a 24-well Aggre-Well™800 plate (Fig 3C). Cocultured spheroids exhibited a significantly different incorporation of NFs and CAFs in DU145 spheroid formation. We observed a higher number of NFs compared to CAFs in each spheroid (Fig 3D). This is likely due to the high proliferation rate characteristic of NFs seen in cell culture (~ 22 hr) compared to CAFs (~38 hr). Although proliferation rates differ, highly uniform and reproducible spheroids were produced suggesting that the AggreWell™800 plate is an effective high-throughput method for producing consistent sizes and shapes of DU145 spheroids monocultured and cocultured with fibroblasts that are uniform within and between experiments.

Using brightfield and confocal microscopy, we observed the spheroid formation of PC3 mono- and cocultures (Fig 4A). PC3 mono-cultured spheroids formed more loose cell aggregates than spheroids, with diameter sizes ranging from of 180 to 490 μm (mean = 271.3 μm) suggesting lower expression of intracellular adhesion molecules by this cell line. However, PC:NF and PC:CAF cocultured spheroids formed compact spheroids with consistent incorporation of fibroblasts and an average diameter size of 366 and 382 μm, respectively (Fig 4B). These PC3, PC:NF, and PC:CAF spheroids showed an average concentration of 527, 319, and 291 spheroids/mL, which slightly exceeds the hypothetical yield due to the loose aggregate formation characteristic of this cell line (Fig 4C). Cocultured spheroids displayed a significantly different incorporation of NFs and CAFs in PC3 spheroid formation as well. We observed a higher number of NFs compared to CAFs in each spheroid as seen in DU145 spheroids (Fig 4D). Although monocultured PC3 spheroids differed in diameter and concentration, highly uniform and reproducible PC:NF and PC:CAF spheroids were produced with the same reproducibility as DU145 cocultures.

## DU145 spheroids are sensitive to TRAIL-mediated apoptosis via taxanes despite reduced death receptor expression

To determine whether 3D spheroids are more resistant to taxane plus TRAIL therapy, DU145 cells were cultured in AggreWell™800 plates to make spheroids. DU145 cells were resistant to TRAIL alone as monolayers and maintain TRAIL resistance when cultured as spheroids. DU145 spheroids also maintain the same resistance to CBZ and DTX alone when compared to 2D after 24 hr exposure (Fig 5A). Bright field images show that monolayer cells treated with 100 ng/mL of TRAIL exhibited little to no morphological changes characteristic of apoptosis. Bright field images also indicated that monolayer cells treated with 0.25 μM CBZ and 0.25 μM DTX showed mitotic arrest characteristic of rounded and partially suspended cells. In combination images, cell morphology is characterized by both apoptosis and mitotic arrest. In brightfield images of spheroids, differences in morphology are harder to detect in this 3D cell culture environment. However, when pretreated with 1 μM CBZ and 1 μM DTX and exposed to 400 ng/mL TRAIL, spheroid morphology changes to a more monolayer conditions as cells are disrupted to single cells or small cell aggregates (Fig 5A).

Flow cytometry for apoptosis detection showed that DU145 spheroids display a similar decrease in cell viability when compared to 2D cell cultures as a monolayer after the combination therapy (Fig 5B). Average cell viability for monolayers exposed to CBZ and DTX followed by TRAIL were 22.9% and 24.6%, respectively. Average cell viability for 3D spheroids exposed to CBZ and DTX followed by TRAIL were 36.9 and 24.5%, respectively (Fig 5C). CBZ + TRAIL treated spheroids showed a significantly higher cell viability (14%) than its 2D counterpart. Furthermore, taxane and TRAIL exerted a synergistic inhibitory effect (Q >1.15) in both 2D and 3D cell cultures as evaluated by Jin's formula. Such synergy was observed when 100

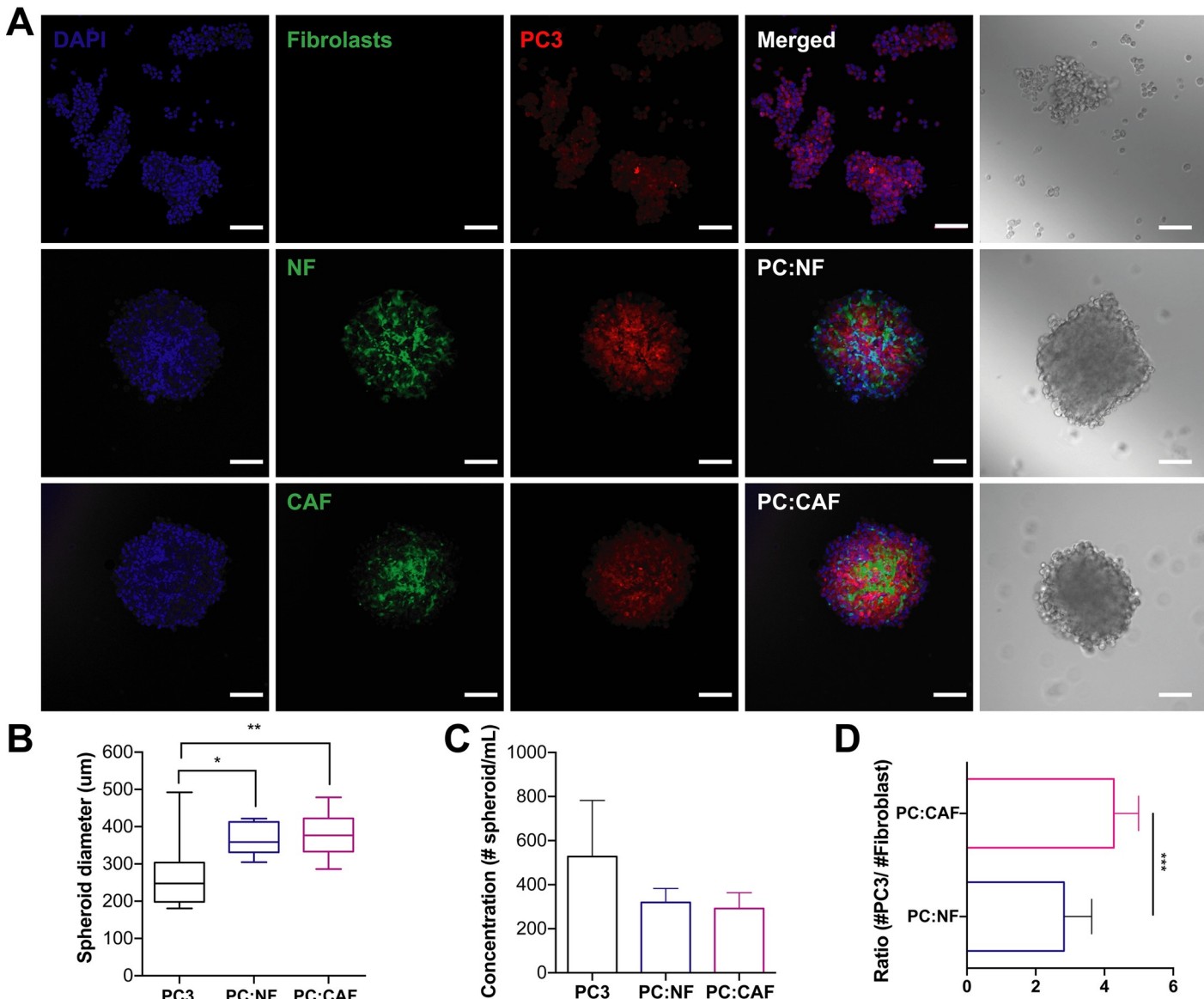

**Fig 4. PC3 spheroids cocultured with prostate fibroblasts in AggreWell™800 plates yielded consistent and reproducible spheroids. A**. Brightfield image and confocal images of PC3 spheroids after 48 hr incubation. Scale bar = 100 μm. **B**. Diameters of PC3, PC3-NF, and PC3-CAF spheroids. **C**. Concentration of spheroids/ mL in PC3, PC3-NF, and PC3-CAF cultures. **D**. Ratio of PC3 cells to fibroblasts in cocultured spheroids. The values represent the mean ± SD (n = 3). * p < 0.05, p < 0.01, **** p < 0.0001.

ng/mL TRAIL was combined with 0.25 μM of either taxane in 2D culture and when 400 ng/ mL TRAIL was combined with 1 μM in 3D culture when compared to DMSO and TRAIL combination (Fig 5D). These data confirm that a synergistic effect is observed via taxane sensitization to TRAIL-induced apoptosis and is promoted in DU145 monolayer cells and spheroids.

We hypothesized sensitivity in DU145 cells would change by reduced death receptor expression when cultured as 3D spheroids. DU145 spheroids expressed lower levels of death receptors DR4 and DR5 in comparison to monolayer cells (Fig 5E). Western blot analysis of whole cell lysate confirmed the decrease in death receptor expression in all 3D spheroid

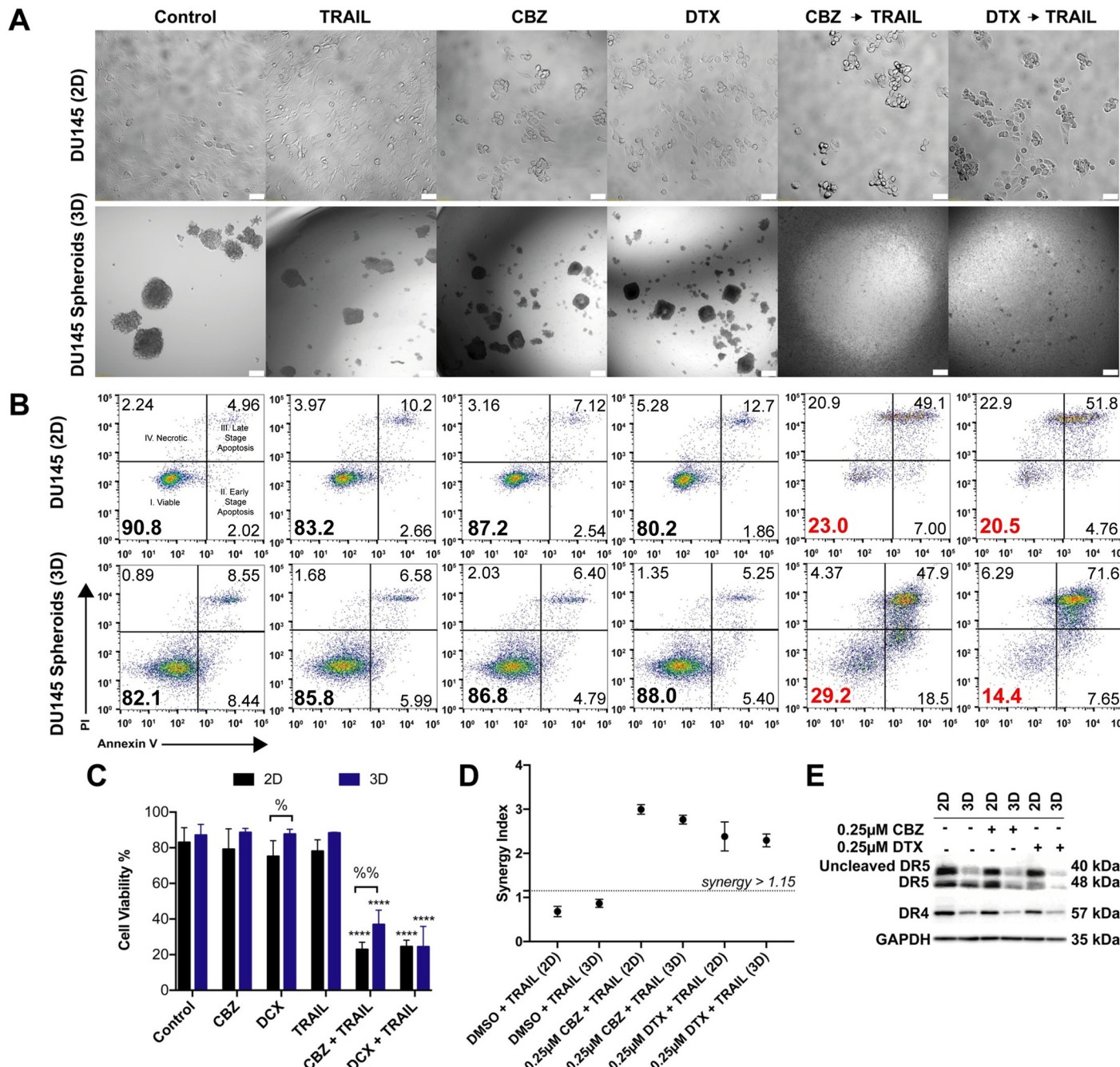

**Fig 5. DU145 spheroids have a lower expression of death receptors but equal sensitivity to taxane plus TRAIL combination therapy. A**. Brightfield images of DU145 cells cultures as 2D monolayer and 3D spheroids after 24 hr treatment exposure with DMSO, TRAIL, CBZ, DTX, CBZ + TRAIL, DTX + TRAIL. Scale bar = 50 μm (top row) and 100μm (bottom row). **B**. Representative flow cytometry plots of 2D and 3D DU145 cells indicating apoptosis based on treatment conditions. **C**. Annexin-V/PI assay results quantifying viability of 2D and 3D DU145 cells under different treatment conditions (n = 6). **D**. Graph displaying synergistic anti-tumor effect of combined taxane and TRAIL in DU145 cells by Jin's formula (n = 3). **E**. Western blot of death receptor expression in 2D and 3D DU145 cells treated with DMSO, CBZ or DTX. The values represent the mean ± SD. **** $p \leq 0.0001$, significantly different from control, % $p < 0.05$, %% $p \leq 0.01$, significantly different 2D vs 3D cultures.

treatment groups. Although death receptor expression decreased, sensitivity to TRAIL treatment post taxane exposure still yielded similar cell viability results when compared to 2D monolayer culture.

## PC3 spheroids are more resistant to TRAIL-mediated apoptosis despite taxanes pretreatment

To determine whether 3D spheroids are more resistant to taxane plus TRAIL therapy, PC3 cells were cultured in AggreWell™800 plates to make spheroids. PC3 cells were resistant to TRAIL alone as monolayers and become more resistant when cultured as spheroids. PC3 spheroids also maintain the same resistance to CBZ and DTX alone when compared to 2D after 24 hr exposure. Bright field images showed that monolayer cells treated with 100 ng/mL of TRAIL exhibited little to no morphological changes characteristic of apoptosis. Bright field images also indicated that monolayer cells treated with 0.25 μM CBZ and 0.25 μM DTX showed mitotic arrest characteristic of rounded and partially suspended cells. In combination images, cell morphology is characterized by both apoptosis and mitotic arrest. In brightfield images of spheroids, differences in morphology are harder to detect in this 3D cell culture environment. However, when pretreated with 1 μM CBZ and 1 μM DTX and exposed to 400 ng/mL TRAIL, spheroid morphology changes to a more 2D-like conditions as cells are disrupted to single cells or small cell aggregates (Fig 6A).

Flow cytometry for apoptosis detection showed that PC3 following combination treatment (Fig 6B). Average cell viability for monolayers exposed to CBZ and DTX followed by TRAIL treatment were 24.3% and 14.1%, respectively. Average cell viability for 3D spheroids exposed to CBZ and DTX followed by TRAIL treatment were 72.5 and 67.2%, respectively (Fig 6C). Furthermore, taxane and TRAIL exerted a synergistic inhibitory effect (Q >1.15) in 2D monolayer cultures when compared to DMSO and TRAIL control combination as evaluated by Jin's formula. The combination therapy exerted a smaller synergistic inhibitory effect of Q = 1.275 in the CBZ + TRAIL treated spheroids. The combination of DTX + TRAIL treated spheroids exhibited an additive effect of Q = 0.819 in 3D cell cultures (Fig 6D). These data suggest that PC3 spheroids are more resistant to TRAIL-induced apoptosis via taxane sensitization as confirmed by the smaller synergistic and additive effect observed.

We hypothesized that the TRAIL sensitivity in PC3 cells would change due to reduced death receptor expression when cultured as 3D spheroids. PC3 spheroids were found to express lower levels of death receptors DR4 and DR5 in comparison to monolayer cells (Fig 6E). Western blot analysis of whole cell lysate confirmed the decrease in death receptor expression in all 3D spheroid treatment groups. With death receptor expression decreasing, sensitivity to TRAIL treatment following taxane exposure still yielded higher cell viability results when compared to 2D monolayer culture. Due to the higher cell viability seen in PC3 spheroids, we determined that cancer stem cells comprised a larger fraction of the total and surviving fraction of PC3 cells. We used flow cytometry to help characterize the surviving fraction of stem-like tumor cells called cancer stem cells (CSC) in the taxane plus TRAIL treated groups. We identified the cancer cells with stem-like characteristics in DU145 and PC3 cells as the CD44 +/CD24- subpopulation (Fig 7A and 7B). In the total cancer cell population of PC3 cells, the CSC subpopulation is moderately higher at ~35% in the control group and remains at this percentage even in both combination treatment groups. Notably, the CSC subpopulation in the viable cell subpopulation was relatively high at ~65% in all groups except the combination groups in which there was a 10% decrease (Fig 7B). A significant difference was observed between the total population and the viable subpopulation in each group, but none was

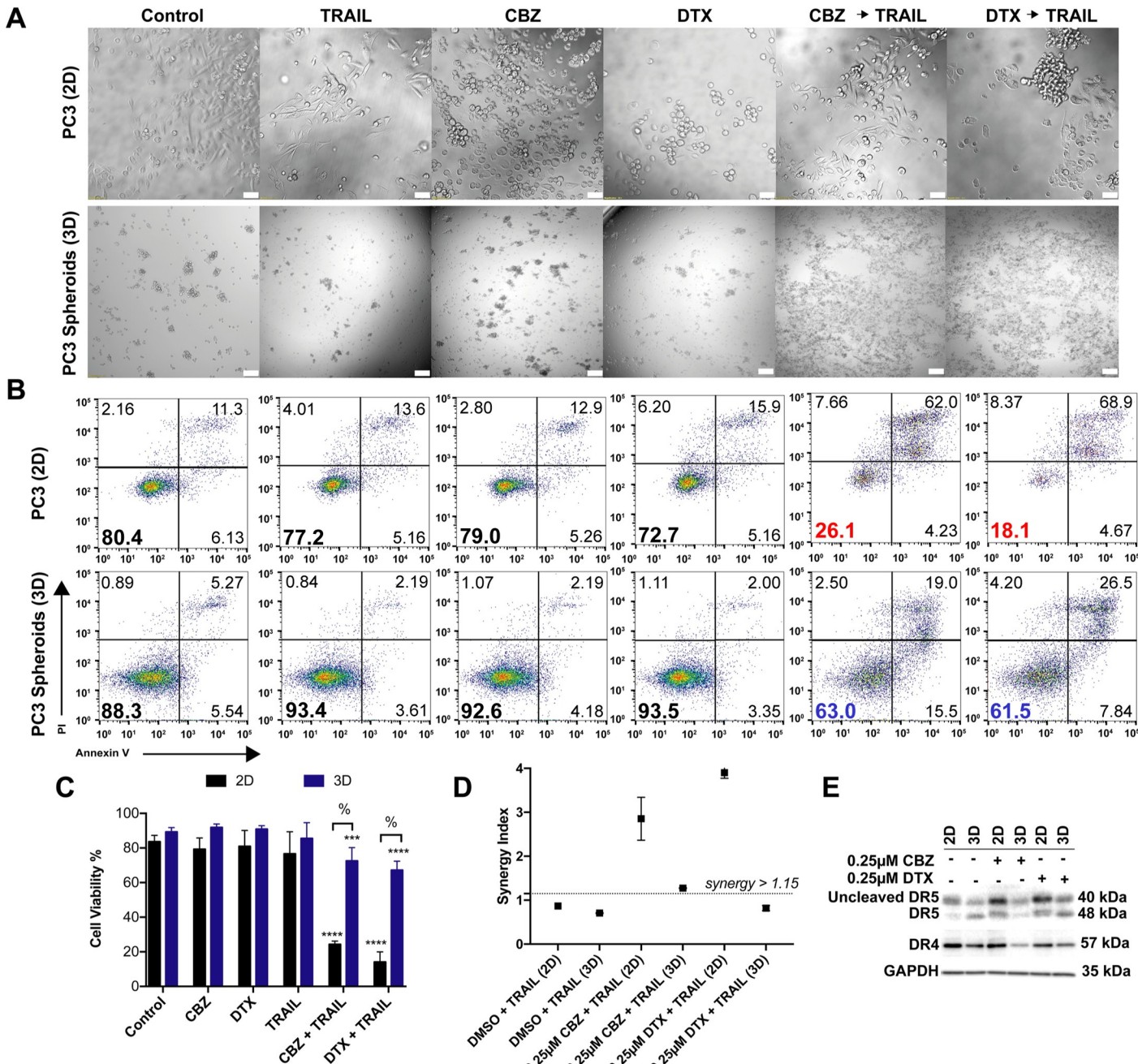

**Fig 6. PC3 spheroids show a lower expression of death receptors and are more resistant to taxane plus TRAIL combination therapy. A**. Brightfield images of PC3 cells cultured as 2D monolayer and 3D spheroids after 24 hr treatment exposure with DMSO, TRAIL, CBZ, DTX, CBZ + TRAIL, DTX + TRAIL. Scale bar = 50 μm (top row) and 100μm (bottom row). **B**. Representative flow cytometry plots of 2D and 3D PC3 cells indicating apoptosis based on treatment conditions. **C**. Annexin-V/ PI assay results quantifying viability of 2D and 3D PC3 cells under different treatment conditions (n = 6). **D**. Graph displaying synergistic anti-tumor effect of combined taxane and TRAIL in PC3 cells by Jin's formula (n = 3). **E**. Western blot of death receptor expression in 2D and 3D PC3 cells treated with DMSO, CBZ or DTX. The values represent the mean ± SD. *** p < 0.001, **** p ≤ 0.0001, significantly different from control, % p < 0.0001, significantly different 2D vs 3D cultures.

observed when comparing each population to their respective controls except for the viable combination group (Fig 7B). The difference in CSCs percentage between to DU145 and PC3 cells can account for the difference in response of PC3 spheroids to the combination treatment.

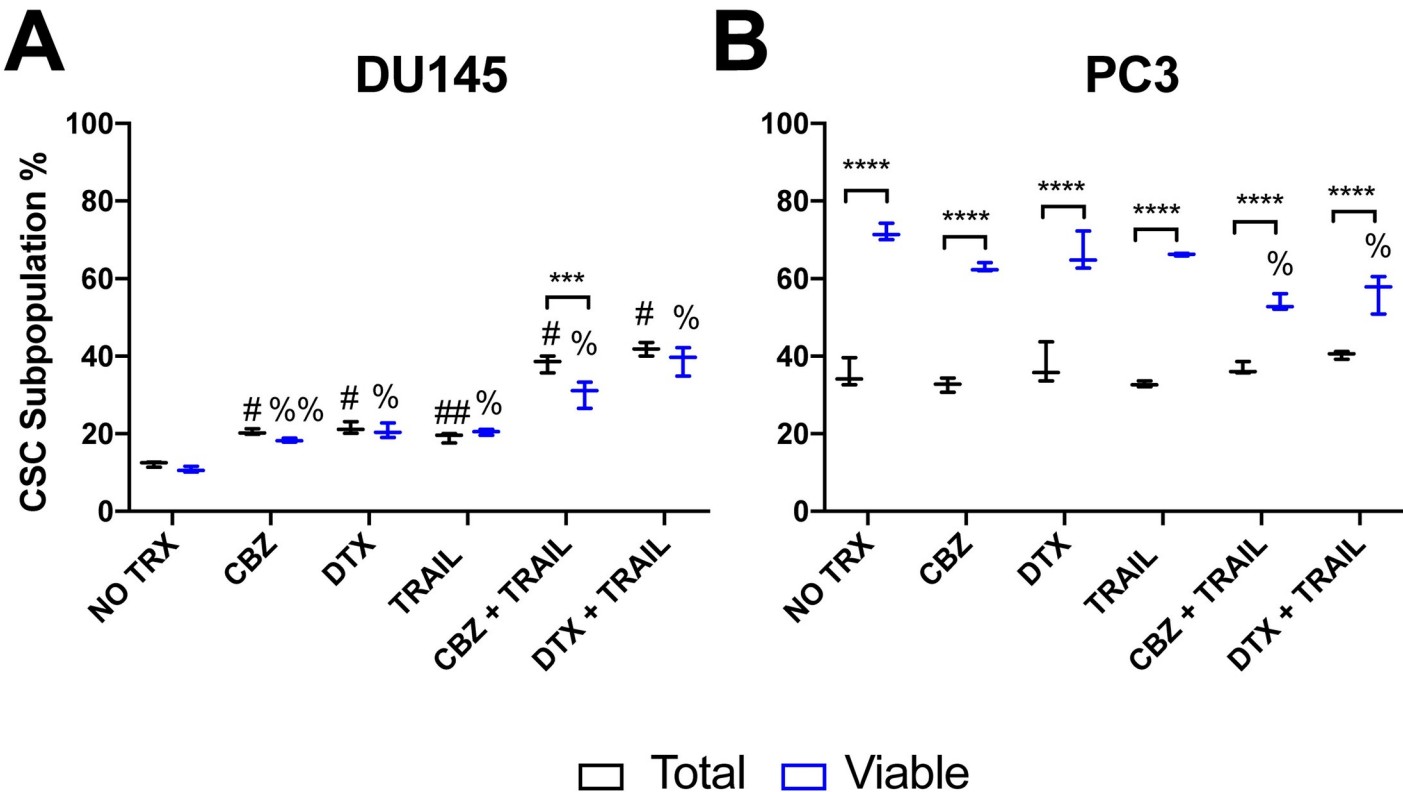

**Fig 7. Analysis of CD44+/CD22- stem cell population in theviable percentage of DU145 and PC3 cells. A, B**. Mean fluorescence intensity of CD44+ and CD24 + cell populations in DU145 and PC3 comparing total and viable populations using Annexin V/PI assay. The values repres ent the mean ± SD (n = 6). *** p < 0.001, **** p ≤ 0.0001, significantly different from total vs viable. ## p ≤ 0.001, # p ≤ 0.0001, significantly different from the total population control. %% p ≤ 0.001, % p ≤ 0.0001, significantly different from the viable supopulation control.

## CAFs and NFs do not confer resistance against taxane plus TRAIL therapy

To investigate the role of prostatic fibroblasts in the chemoresistance of mCRPC spheroids, DU145 and PC3 spheroids cocultured with NFs and CAFs were treated with taxanes to compare sensitization effects. Spheroid diameter for DU145 and PC3 spheroid co-cultures were similar in size. DU145 and PC3 monocultured spheroids showed a significant difference in diameter due to PC3 cells forming loose cell aggregates (Fig 8A). However, the spheroid concentration for all spheroids was not significantly different (Fig 8B). NFs in DU:NF and PC:NF spheroids displayed similar ratios, but CAFs in DU:CAF and PC:CAF spheroids displayed higher ratios that were significantly different. These data suggests that less CAFs where being incorporated because of differing proliferation rates (Fig 8C).

Mono- and cocultured spheroids were treated with 1 μM CBZ and 1 μM DTX and were subsequently exposed to 400 ng/mL TRAIL. Flow cytometry analysis revealed that combined taxane and TRAIL treatment significantly reduced the viability of both DU145 mono- and cocultured spheroids. Cell viability for all three culture conditions when treated with taxane or trail alone was > 80%. When treated with CBZ and TRAIL, average cell viability was dramatically reduced to 36.9%, 30.6%, and 27.2% for DU145, DU:NF, DU:CAF spheroids, respectively. When treated with DTX and TRAIL, average cell viability dropped to 24.5%, 26.1%, and 34.5% for DU145, DU:NF, DU:CAF spheroids, respectively (Fig 8D). No significant difference was observed in either combination treatment group suggesting neither NFs nor CAFs confer additional resistance to the combination treatment in DU145 spheroids.

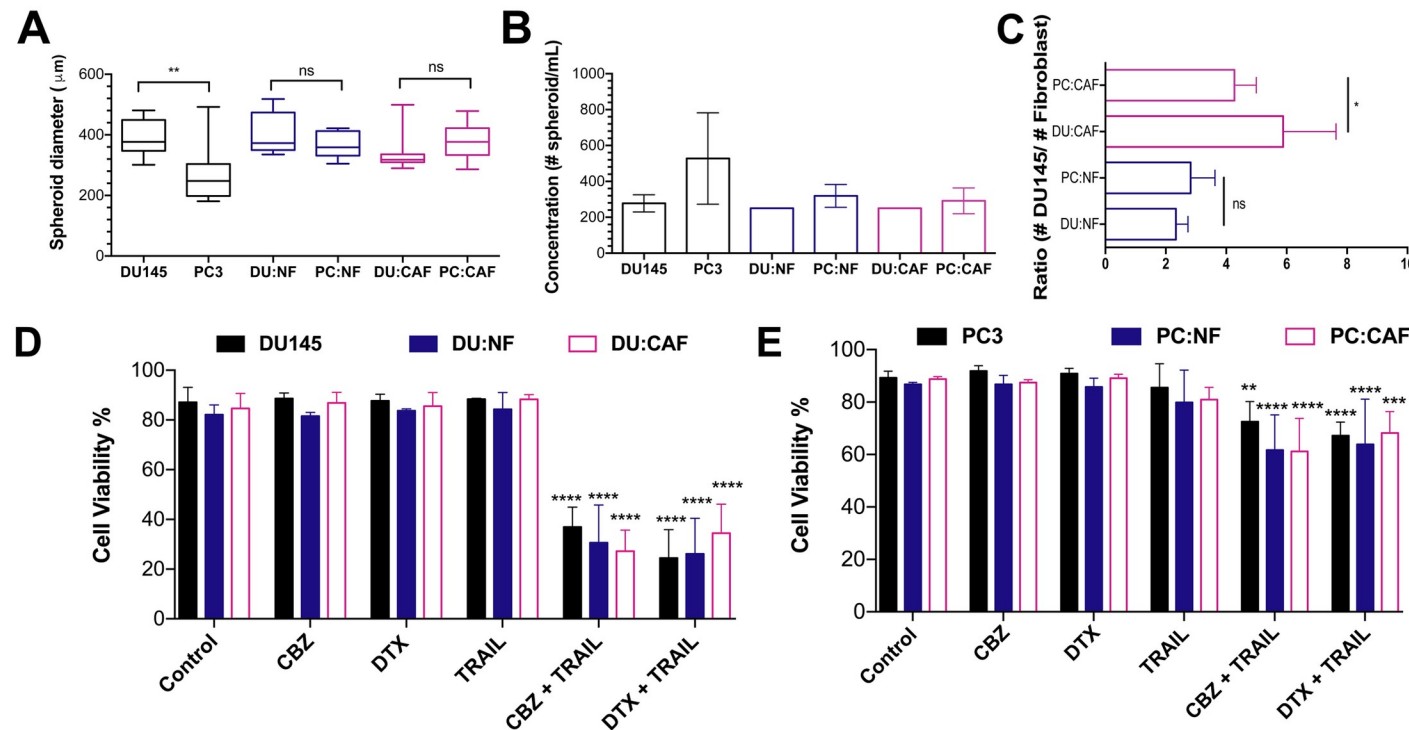

**Fig 8. Cocultured spheroids show similar drug resistance to taxane plus TRAIL therapy. A**. Diameters of mono- and cocultured DU145 and PC3 spheroids with NFs and CAFs. **B**. Concentration of spheroids/mL of mono- and cocultured DU145 and PC3 spheroids with NFs and CAFs. **C**. Ratio of NFs and CAFs in co-cultured DU145 and PC3 spheroids. **D**. Ratio of PC3 cells to fibroblasts in cocultured spheroids. The values represent the mean ± SD (n = 3). * p < 0.05, p ** < 0.01, **** p < 0.0001, significantly different from control.

For PC3 spheroids, flow cytometry analysis revealed that combined taxane and TRAIL treatment significantly reduced the viability of both mono- and cocultured spheroids; however, viability only decreased ~20% compared to ~50% for DU145 spheroids. Cell viability for all three culture conditions when treated with taxane or TRAIL alone was also > 80%. When treated with CBZ and TRAIL, average cell viability dropped to 72.5%, 61.7%, and 61.2% for PC3, PC:NF, PC:CAF spheroids, respectively. When treated with DTX and TRAIL, average cell viability dropped to 67.1%, 63.8%, and 68.2% for PC3, PC:NF, PC:CAF spheroids, respectively (Fig 8C). Similar to DU145 spheroids, neither NFs nor CAFs confer additional resistance to the combination treatment suggesting that taxane plus TRAIL may also sensitize fibroblasts to TRAIL-induced apoptosis.

## Discussion/Conclusions

Three-dimensional tumor models recapitulate the heterogenic organization and complex intracellular network of the TME. Traditional 2D cell cultures lack this complexity and fail to reproduce characteristics of tumors resulting in differential therapy responses that falter *in vivo* and do not pass the pre-clinical phase [5, 6]. Here, we compared 2D monolayers to 3D monocultured PCa spheroids. Using AggreWell™800 microwell plates, we were able to make consistent and highly reproducible DU145 and PC3 spheroids in large quantities for TRAIL sensitization studies using taxanes, CBZ and DTX. Taxanes stabilize microtubules in cells, inhibiting microtubule depolymerization, arresting cells in $G_2$ and M phase, resulting in cell death [15]. Docetaxel is the first-line chemotherapeutic option given to patients with mCRPC [14]. Unfortunately, patients eventually develop resistance to this paclitaxel derivative, limiting

patient survival [15]. Cabazitaxel is a more potent third-generation taxane that was developed to overcome docetaxel-resistant cells [28]. There are several studies that indicate the utility of cabazitaxel as a first-line treatment in mCRPC due to improved cytotoxicity and overall survival [28]. In this study we used docetaxel and cabazitaxel as first-line treatments for the sensitization of PCa cells to TRAIL.

DU145 and PC3 cell lines are of interest because they are androgen-independent and represent mCRPC *in vitro*. Both cell lines have been shown to be resistant to TRAIL and studies have been performed to identify ways to sensitize them to TRAIL [29–32]. Consequently, we investigated if cells cultured as 3D spheroids are more resistant to TRAIL-mediated apoptosis via taxane treatment than 2D monolayers (Figs 5 and 6). There is growing evidence that the upregulation of DR5 is a pathway to sensitizing tumor cells to TRAIL-induced apoptosis (Figs 1 and 2). A possible explanation is that taxane exposure induces DR5 transcription expression, thereby increasing protein availability at the cell surface. Overall, DU145 spheroids showed a lower expression of DR4/5 when compared to 2D; however, DU145 spheroids still exhibited significant apoptosis similar to 2D when treated with the combination therapy. We believe that taxane pretreatment makes TRAIL more potent in this cell line by other mechanisms such as ER stress. PC3 spheroids also had a lower expression of DR4/5 when compared to 2D monolayers. PC3 spheroids were more resistant to taxane plus TRAIL therapy and showed an increase in viability compared to cells cultured as a monolayer. This significant increase in cell viability is most likely due to a high percentage of cancer stem cells present even in 2D PC3 cell culture (Fig 7). Tumor spheroids become enriched with cells exhibiting cancer stem cell phenotypes that are more resistant to chemotherapies [2]. The difference in responses in PC3 cells in 2D and 3D culture, unlike DU145, highlight a need to tailor therapies in a patient-specific and cell-specific manner to tackle the cell heterogeneity found in solid tumors of mCRPC.

Additionally, CAFs are the most prominent cell type within the tumor stroma of many cancers, including prostate cancer [33]. CAFs express more TGF-B and fibronectin which regulate proliferation and migration compared to NF [34, 35]. There is also differential expression of cytokines and fibroblast-derived factors released by NFs and CAFs that play a role in conferring resistance to shear forces between CAFs and NFs [25]. CAFs have been shown to positively induce tumorigenicity and CSCs contributing to the invasiveness of PCa [36–39]. We cocultured PCa cells and fibroblasts to mimic stromal-cancer cell interactions and chemotherapy resistance seen in mCRPC. We found that CAFs and NFs formed compact DU145 spheroids (Fig 3). PC3 spheroids formed loose aggregates when monocultured but cocultures with NFs and CAFs enhanced spheroid formation (Fig 4). This enhanced formation may be influenced by the intercellular adhesion molecules, matrix metalloproteinases, and collagen expressed by fibroblasts [40, 41]. Neither NFs nor CAFs seemed to enhance resistance when compared to monocultured spheroids, suggesting that other mechanisms and cell types are responsible for TRAIL sensitivity or resistance depending on the cell line (Fig 8). CAFs play a more active role in conserving the proliferative ability of tumor cells and inducing resistance when under high magnitude fluid shear stress, unlike the low shear stress environment of a typical solid tumor [25]. Multicellular tumor spheroids like these are key in predicting *in vivo* efficacy of various chemotherapies and combination therapies to determine patient response [42–45]. This highly reproducible method allows for more efficient characterization of drug testing and efficacy in spheroid platforms that can help bridge the gap between i*n vitro* and *in vivo* experiments allowing for more patient-specific therapies to effectively treat mCRPC.

## Supporting information

**S1 Fig. Original images for Western blots and gels.**
(PDF)

**S2 Fig. Mycoplasma test results via Mycoplasma detection kit (ATCC 30-1012K).**
(PDF)

## Author Contributions

**Conceptualization:** Korie A. Grayson.

**Formal analysis:** Korie A. Grayson.

**Funding acquisition:** Michael R. King.

**Investigation:** Korie A. Grayson, Nidhi Jyotsana, Nerymar Ortiz-Otero.

**Methodology:** Korie A. Grayson, Nerymar Ortiz-Otero.

**Project administration:** Michael R. King.

**Supervision:** Michael R. King.

**Writing – original draft:** Korie A. Grayson.

**Writing – review & editing:** Korie A. Grayson, Michael R. King.

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
