## [Decision Letter · Decision Letter 0]

5 Jul 2020

PONE-D-20-14145

Overcoming TRAIL-resistance by sensitizing prostate cancer 3D spheroids with taxanes

PLOS ONE

Dear Dr. King,

Thank you for submitting your manuscript to PLOS ONE. After careful consideration, we feel that it has merit but does not fully meet PLOS ONE’s publication criteria as it currently stands. Therefore, we invite you to submit a revised version of the manuscript that addresses the points raised during the review process.

Additional experiments are required to confirm authors' conclusions.

More detailed discussion is needed for the results presented in the manuscript

We look forward to receiving your revised manuscript.

Kind regards,

Irina V. Lebedeva, Ph.D.

Academic Editor

PLOS ONE

Journal Requirements:

2. Please provide additional information about each of the cell lines used in this work, including any quality control testing procedures (authentication, characterisation, and mycoplasma testing). For more information, please see http://journals.plos.org/plosone/s/submission-guidelines#loc-cell-lines.

3.PLOS ONE now requires that authors provide the original uncropped and unadjusted images underlying all blot or gel results reported in a submission’s figures or Supporting Information files. This policy and the journal’s other requirements for blot/gel reporting and figure preparation are described in detail at https://journals.plos.org/plosone/s/figures#loc-blot-and-gel-reporting-requirements and https://journals.plos.org/plosone/s/figures#loc-preparing-figures-from-image-files. When you submit your revised manuscript, please ensure that your figures adhere fully to these guidelines and provide the original underlying images for all blot or gel data reported in your submission. See the following link for instructions on providing the original image data: https://journals.plos.org/plosone/s/figures#loc-original-images-for-blots-and-gels.

4.PLOS requires an ORCID iD for the corresponding author in Editorial Manager on papers submitted after December 6th, 2016. Please ensure that you have an ORCID iD and that it is validated in Editorial Manager. To do this, go to ‘Update my Information’ (in the upper left-hand corner of the main menu), and click on the Fetch/Validate link next to the ORCID field. This will take you to the ORCID site and allow you to create a new iD or authenticate a pre-existing iD in Editorial Manager. Please see the following video for instructions on linking an ORCID iD to your Editorial Manager account: https://www.youtube.com/watch?v=_xcclfuvtxQ

Reviewers' comments:

Reviewer's Responses to Questions

**Comments to the Author**

1. Is the manuscript technically sound, and do the data support the conclusions?

Reviewer #1: Yes

Reviewer #2: Partly

Reviewer #3: Partly

2. Has the statistical analysis been performed appropriately and rigorously? 

Reviewer #1: Yes

Reviewer #2: Yes

Reviewer #3: No

3. Have the authors made all data underlying the findings in their manuscript fully available?

Reviewer #1: Yes

Reviewer #2: Yes

Reviewer #3: No

4. Is the manuscript presented in an intelligible fashion and written in standard English?

Reviewer #1: Yes

Reviewer #2: Yes

Reviewer #3: Yes

5. Review Comments to the Author

Reviewer #1: In this manuscript, the authors compare sensitivity of DU145 and PC3 prostate cancer cell lines grown in 2D vs 3D to the

Please provide additional details regarding the CAFs used (hTERT PF179T). Were these purchased or created in the authors’ lab?

Figures 1, 2: Does the ratio of prostate cancer cells to fibroblasts used to create the spheroids impact the ratio of each or the size of the final formed spheroid? This should be directed addressed.

Page 12, Lines 260-270: results from several experiments are described but this data is not shown in any of the figures and quantitative results are not provided. If these results are to be included in the manuscript, the data should be included in the figures and a quantitative measure of outcomes (cell death, sensitivity, etc) should be provided with appropriate statistics.

Page 12: Line 260 states that DU145 cells were resistant to TRAIL alone, but then Line 263-264 state that treated DU145 cells show morphological changes of apoptosis. This seems at odds. This should be clarified and the data and analysis should be provided.

Use of the word “cells” can be confusing in places because its variably used to refer to cells in 2D vs 3D. Please try to clarify; for example, could refer to 3D cultures strictly as “spheroids”, or always referring to 2D cultures as “monolayer cells”.

Page 14, Line 301: Typo…I think “DU145” should be “PC3”

Page 14, Lines ~300-310: Same issue as in preceding DU145 section. Authors state that treatment did not impact cells, but then describe morphological changes of apoptosis…this seems discrepant.?

The difference in sensitivity to the TRAIL+chemo combo in 2D vs 3D of DU145 (no difference) and PC (big difference) is perhaps the most interesting result from Figures 3-4 and this difference in behavior between the two cell lines could be emphasized.

For experiments in Figure 5, how can death of cancer cells be differentiated from death of fibroblasts following treatment? This should be explained. If the assay is not able to differentiate death of cancer cells from fibroblasts, this could impact interpretation of results.

To my knowledge, there is little available clinical data to support targeting TRAIL in prostate cancer. If such data is available, it would be worthwhile to discuss since it is otherwise not so clear why the authors chose to focus on TRAIL therapy in combination with taxanes (which do have a demonstrated role in clinical management of mCRPC).

Reviewer #2: The authors have evaluated the effects of TRAIL, cabazitaxel and docetaxel in 3D spheroids. The results of this paper show some benefits of combination therapy in vitro. The endpoints of the study should be better described. More experimental work is needed to completely support the conclusions. This reviewer has the following specific suggestions and recommendations:

1. Page 12, the differences between NAF and CAF are mentioned. What is the difference in expression of key molecules which regulate proliferation and migration between NAF and CAF? This is not clearly explained in the manuscript.

2. apoptosis is detected mostly by morphological changes. This is not sufficient. Additional tests should be considered. For example, the authors should use PARP cleavage. This is an important part of the manuscript and should be better presented.

3. In order to further support the findings on page 14, I recommend to perform the cell cycle analysis.

4. in order to ensure consistency in the manuscript, I recommend to analyze the expression of death receptors (DR) after combination therapy (page 17).

5. Importantly, the argumentation for use of only androgen receptor-negative cell lines in the paper is not complete. Key findings should be repeated with at least one androgen receptor-positive cell line.

Reviewer #3: This manuscript seeks to test the hypothesis that the combination of TRAIL + the taxanes docetaxel (DTX) and cabazitaxel (CBZ) sensitizes the prostate cancer cell lines PC3 and DU145 to cell death compared to treatment with these agents alone. While the article has several interesting points it has major issues that would require careful attention and further experimentation.

1. The authors part from the premise that 2D and 3D cultures of PC3 and DU145 cells are resistant to treatment with DTX or CBZ because no apoptosis is observed after 24 h of treatment with 0.25 uM (250 nM) of each drug. There are two major problems with this rationale:

a) Treatment for 24 h, according to the Methods and Results sections, is not sufficient time to induce PC3 or DU145 apoptotic cell death with taxanes. The reason is that for taxanes to induce apoptosis, the cells need to be first arrested in mitosis, which is then followed up by cells rounding up, detaching from the plates, and undergoing mitotic catastrophe. For significant cell death to be observed, this process needs to be followed up for approximately 48-72 h.

b) 0.25 uM, or 250 nM, is a very high concentration of taxanes, which may not induce apoptotic cell death at 24 h for the reasons stated above but may induce lysosomal or other non-apoptotic modes of cell death after 24 h. Typically PC3 and DU145 cells growing in 2D are sensitive and undergo apoptotic cell death when treated a low nM concentrations (5-20 nM) for 48 h - 72 h. In fact, DTX-resistant PC3 and DU145 cells are normally selected and maintained in culture at 10 nM concentrations. The use of very high taxane concentrations in this manuscript must be justified. The claim that these cells are resistant to high taxane concentrations after treatment for only 24 h is questionable. The cells are simply not ready to display the characteristics of apoptosis at that time point. The authors should use a dose-response, time-response experiment to compare taxane resistance between 2D or 3D PC3 and DU145 cell lines, or select the cells for taxane resistance for several weeks by incrementally increasing drug concentrations while selecting surviving cells.

2. The combination treatment of taxane plus TRAIL at 48 h does induce cell death, but the problem is that apparently the treatments with taxanes alone and TRAIL alone were conducted for 24 h. This is what the Methods and Results sections imply, even when the figure legends suggest that treatments were done at 48 h. This needs to be clarified because it is possible that the observed loss of viability could be due to induction of cell death by taxanes after 48 incubation, and not by the TRAIL-taxane combination. If the single treatments were done at 24 h and the combinatorial treatment at 48 h, then the following comparisons are needed: single and dual treatments at both 24 h and 48 h.

3. Was TRAIL used as 400 ng/ml or 100 ng/ml? Both concentrations are mentioned in the text without explanation.

4. The measurement of apoptosis using the Annexin V/PI method needs a positive control to rule out the possibility that trypsinization may have cleaved Annexin V from the cell surface, which may have resulted in underestimation of cell death. The combination of TRAIL (100 ng/ml) + Actinomycin D causes robust apoptosis between 6-12 h and could be used as a suitable positive control for this assay.

5. The flow cytometry panels in Figures 3 and 4 need better resolution. Very difficult to see the numbers.

6. Was necrotic cell death estimated (AV-/PI+) for each treatment? If so the numbers should be provided more clearly in the flow figures. It is possible that the high taxane concentrations may have induced some necrotic cell death.

7. Treatment with DTX or CBZ led to downregulation of the TRAIL receptors DR4 and DR5 in both PC3 and DU145 spheroids; however, 3D DU145 cultures were still sensitive to TRAIL + taxanes whereas PC3 cultures were more resistant, suggesting that downregulation of these receptors may not be related to the cell viability response. The authors argue that this difference may be related to the presence of more stem cells in the PC3 spheroids but the data is not presented. These contradictory results need more detailed discussion, and perhaps additional supportive data.

8. To establish the role of the DR4 and DR5 receptors in influencing sensitivity to TRAIL + taxanes in PC3 and DU145 spheroids the authors should knockdown these receptors in 2D cells and examine culture sensitivity to this combinatorial treatment.

9. The cancer cells and the the fibroblasts were stained with different Cell Tracker fluorescent dyes. How do the authors know that these dyes are specific for cancer cells or fibroblasts?

10. It is not clear from the Methods if the PCa cells were mixed with the fibroblasts and then seeded as a mixture, or if the fibroblasts and the PCa cells were seeded sequentially. Regardless, it seems that the presence of fibroblasts does not influence cellular responses to TRAIL + taxanes. Discussion of this issue needs elaboration.

11. What are the comparison groups in Fig. 5, particularly for the bars with significant values? The statistical analysis needs to be clarified for this figure.

12. There are several key statements in the Introduction and Discussion sections that need citations.

6. PLOS authors have the option to publish the peer review history of their article (what does this mean?). If published, this will include your full peer review and any attached files.

Reviewer #1: No

Reviewer #2: No

Reviewer #3: No

---

## [Author Response · Author response to Decision Letter 0]

31 Dec 2020

Comments to the Author 

Reviewer #1: In this manuscript, the authors compare sensitivity of DU145 and PC3 prostate cancer cell lines grown in 2D vs 3D to the

Please provide additional details regarding the CAFs used (hTERT PF179T). Were these purchased or created in the authors’ lab?

On page 6, lines 128, we stated that the hTERT PF179T CAFs were obtained from American Type Culture Collection. We have changed the word “obtained” to “purchased” to make this more clear. 

Figures 1, 2: Does the ratio of prostate cancer cells to fibroblasts used to create the spheroids impact the ratio of each or the size of the final formed spheroid? This should be directed addressed.

The ratio does not affect size as opposed to the total number of cells which does. More cells result in larger spheroids. We performed some pilot experiments with 3.0x106 cells (twice the number of cells used in this study) and saw that spheroids where larger when more cells were utilized in the culture. We settled on 1.5x106 total number of cells and a 2:1 ratio of cancer cells to fibroblasts due to the larger size of fibroblasts compared to cancer cells. This is addressed in the manuscript on page 15, lines 330-331.

Page 12, Lines 260-270: results from several experiments are described but this data is not shown in any of the figures and quantitative results are not provided. If these results are to be included in the manuscript, the data should be included in the figures and a quantitative measure of outcomes (cell death, sensitivity, etc) should be provided with appropriate statistics.

The current lines 381-393 contain a qualitative description of the results based on brightfield images in Figure 5A. Lines 405-416, Figure 5B, and Figure 5C quantitatively describe the results of each treatment. 

Page 12: Line 260 states that DU145 cells were resistant to TRAIL alone, but then Line 263-264 state that treated DU145 cells show morphological changes of apoptosis. This seems at odds. This should be clarified and the data and analysis should be provided.

We thank the reviewer for pointing out this inconsistency, we previously intended to indicate that there were little to no observable morphological changes characteristic of apoptosis. This description has been edited appropriately on page 17, lines 385-386.

Use of the word “cells” can be confusing in places because its variably used to refer to cells in 2D vs 3D. Please try to clarify; for example, could refer to 3D cultures strictly as “spheroids”, or always referring to 2D cultures as “monolayer cells”.

Terminology for “cells” has been changed to spheroids when referring to 3D cultures and monolayer cells when referring to 2D cultures, as suggested by the reviewer.

Page 14, Line 301: Typo…I think “DU145” should be “PC3”

This error has been corrected, thank you for identifying this.

Page 14, Lines ~300-310: Same issue as in preceding DU145 section. Authors state that treatment did not impact cells, but then describe morphological changes of apoptosis…this seems discrepant.?

We recognized the potential for confusion and previously intended to indicate that there were little to no observable morphological changes characteristic of apoptosis. This text has been edited on page 19, lines 429-430.

The difference in sensitivity to the TRAIL+chemo combo in 2D vs 3D of DU145 (no difference) and PC (big difference) is perhaps the most interesting result from Figures 3-4 and this difference in behavior between the two cell lines could be emphasized.

The difference in behavior between the two cell lines is now emphasized in Figure 7 and on page 21 lines 461-480. The behavior is also discussed in the discussion pertaining to the presence of cancer stem cells that can add to resistance especially in 3D form now found on page 25, lines 559-564. 

For experiments in Figure 5, how can death of cancer cells be differentiated from death of fibroblasts following treatment? This should be explained. If the assay is not able to differentiate death of cancer cells from fibroblasts, this could impact interpretation of results.

This comment concerns the current Figure 8. The authors recognize this important point and specifically stained each cell line with different Cell Tracker colors to enable straightforward identificataion each population via single-cell flow cytometry analysis. The stained fibroblasts where unambiguously differentiated from the cancer cell population due to the major different wavelength of the Cell Tracker dyes. The cancer cell population was selected and then results analyzed in FlowJo. Clarification of the staining and assay has been made in the methods section under “Apoptosis Assay” on page 8, lines 179-180.

To my knowledge, there is little available clinical data to support targeting TRAIL in prostate cancer. If such data is available, it would be worthwhile to discuss since it is otherwise not so clear why the authors chose to focus on TRAIL therapy in combination with taxanes (which do have a demonstrated role in clinical management of mCRPC).

We focused on TRAIL because there are previous data showing that resistance to TRAIL in prostate cancer cells can be overcome by treatment with small molecules and chemotherapeutic agents that is detailed in a separate manuscript currently in review elsewhere. We addressed this on page 4, lines 85-86. 

Reviewer #2: The authors have evaluated the effects of TRAIL, cabazitaxel and docetaxel in 3D spheroids. The results of this paper show some benefits of combination therapy in vitro. The endpoints of the study should be better described. More experimental work is needed to completely support the conclusions. This reviewer has the following specific suggestions and recommendations:

1. Page 12, the differences between NAF and CAF are mentioned. What is the difference in expression of key molecules which regulate proliferation and migration between NAF and CAF? This is not clearly explained in the manuscript.

CAFs express more TGF-B (Life Sciences, 2019) and fibronectin (J Cell Biol. 2017) which regulate proliferation and migration compared to NF. There is also an upregulation of soluble factors that play a role in conferring resistance to shear forces between CAFs and NFs (Oncotarget 2020). This is now explained in the discussion section on page 25, lines 566-570.

2. Apoptosis is detected mostly by morphological changes. This is not sufficient. Additional tests should be considered. For example, the authors should use PARP cleavage. This is an important part of the manuscript and should be better presented.

This description suggesting apoptosis via morphological changes was poorly phrased in our original submission, what we intended to state is that there were little to no observable morphological changes characteristic of apoptosis for the control groups. We did to use Annexin V/PI assay to determine stages of apoptosis as seen in Figures 5B, 5C, 6B, 6C. 

3. In order to further support the findings on page 14, I recommend to perform the cell cycle analysis.

Cell cycle analysis has been performed via a propidium iodide based cell cycle assay as suggested by the reviewer. Results are presented in Figures 1-2 and on pages 13-14 for the DU145, PC3, and LNCaP cell lines. 

4. In order to ensure consistency in the manuscript, I recommend to analyze the expression of death receptors (DR) after combination therapy (page 17).

After combination therapy, TRAIL was observed to bind to the death receptors and initiate apoptosis resulting in over 80% cell death. Analysis of death receptor expression at this stage is not feasible since most cells are dead, and the cell concentration would be too low to detect DR via Western blot. Previous studies (Int J Mol Sci 2017, Can Res 2001) and a study we currently have in review elsewhere have examined changes in DR expression in response to taxane therapy. This is noted on page 13, lines 284-285. 

5. Importantly, the argumentation for use of only androgen receptor-negative cell lines in the paper is not complete. Key findings should be repeated with at least one androgen receptor-positive cell line.

Key findings have been repeated for the androgen-dependent cell line LNCaP (as suggested by the reviewer) and the results have been presented in Figure 2 and on page 14. 

Reviewer #3: This manuscript seeks to test the hypothesis that the combination of TRAIL + the taxanes docetaxel (DTX) and cabazitaxel (CBZ) sensitizes the prostate cancer cell lines PC3 and DU145 to cell death compared to treatment with these agents alone. While the article has several interesting points it has major issues that would require careful attention and further experimentation.

1. The authors part from the premise that 2D and 3D cultures of PC3 and DU145 cells are resistant to treatment with DTX or CBZ because no apoptosis is observed after 24 h of treatment with 0.25 uM (250 nM) of each drug. There are two major problems with this rationale:

a) Treatment for 24 h, according to the Methods and Results sections, is not sufficient time to induce PC3 or DU145 apoptotic cell death with taxanes. The reason is that for taxanes to induce apoptosis, the cells need to be first arrested in mitosis, which is then followed up by cells rounding up, detaching from the plates, and undergoing mitotic catastrophe. For significant cell death to be observed, this process needs to be followed up for approximately 48-72 h.

Our goal of treatment is not to induce apoptotic cell death with taxanes. We chose a taxane concentration and timepoint that would only sensitize prostate cancer cells to TRAIL-induced apoptosis, not taxane-induced apoptosis which is addressed in a separate manuscript that is currently in review elsewhere. We have added clarifying discussion related to this reasoning in the manuscript on page 13, lines 276-277 and have included a new time-response comparing percent apoptosis from 24-96 hr in Figure 1.

b) 0.25 uM, or 250 nM, is a very high concentration of taxanes, which may not induce apoptotic cell death at 24 h for the reasons stated above but may induce lysosomal or other non-apoptotic modes of cell death after 24 h. Typically PC3 and DU145 cells growing in 2D are sensitive and undergo apoptotic cell death when treated a low nM concentrations (5-20 nM) for 48 h - 72 h. In fact, DTX-resistant PC3 and DU145 cells are normally selected and maintained in culture at 10 nM concentrations. The use of very high taxane concentrations in this manuscript must be justified. The claim that these cells are resistant to high taxane concentrations after treatment for only 24 h is questionable. The cells are simply not ready to display the characteristics of apoptosis at that time point. The authors should use a dose-response, time-response experiment to compare taxane resistance between 2D or 3D PC3 and DU145 cell lines, or select the cells for taxane resistance for several weeks by incrementally increasing drug concentrations while selecting surviving cells.

We recognize that 0.25�M is a high concentration of taxane exposure; however we only saw a significant response regarding TRAIL-induced apoptosis at this concentration. Lower concentrations (0.0078-0.125�M) induced only ~50% apoptosis while higher concentrations (0.25-2�M) induced ~80% apoptosis to occur when exposed to TRAIL after 24 hr treatment with taxanes. This concentration is also used in a prior study: (Int J Mol Sci 2017). We discuss our reasoning in the manuscript on page 12, lines 251-258 and now include a new dose-response curve for both cell lines comparing the percent apoptosis for all the concentrations tested as stated above with TRAIL in Figure 1.

2. The combination treatment of taxane plus TRAIL at 48 h does induce cell death, but the problem is that apparently the treatments with taxanes alone and TRAIL alone were conducted for 24 h. This is what the Methods and Results sections imply, even when the figure legends suggest that treatments were done at 48 h. This needs to be clarified because it is possible that the observed loss of viability could be due to induction of cell death by taxanes after 48 incubation, and not by the TRAIL-taxane combination. If the single treatments were done at 24 h and the combinatorial treatment at 48 h, then the following comparisons are needed: single and dual treatments at both 24 h and 48 h.

The combination treatment of 48h = 24 h taxane + 24 h TRAIL. Media containing taxanes at 0.25�M is removed from the well after 24 h and then replaced with media containing 100 ng/mL of TRAIL for 24 more hours. We now clarify this time course in the methods section to avoid confusion. We selected a sequential treatment with taxane and TRAIL because we wanted to determine the ability and capacity of sensitization that occurs from taxane alone. This is now clarified in the methods section and on page 8 and page 9.

3. Was TRAIL used as 400 ng/ml or 100 ng/ml? Both concentrations are mentioned in the text without explanation.

100 ng/mL was used for the 2D cell cultures and 400 ng/mL was used for treatment of the spheroids. The concentration was scaled up by the difference in media volume between 2D and 3D cell cultures. We have added this explanation to page 8, lines 180-181.

4. The measurement of apoptosis using the Annexin V/PI method needs a positive control to rule out the possibility that trypsinization may have cleaved Annexin V from the cell surface, which may have resulted in underestimation of cell death. The combination of TRAIL (100 ng/ml) + Actinomycin D causes robust apoptosis between 6-12 h and could be used as a suitable positive control for this assay.

When lifting the cells to analyze for apoptosis using the Annexin V/PI, trypsin was not used. Accutase� was used for detachment which is much gentler on the cells and is not expected to cleave membrane proteins as trypsin. This has been corrected in the methods section of the manuscript on page 8, lines 184.

5. The flow cytometry panels in Figures 3 and 4 need better resolution. Very difficult to see the numbers.

We have increased the size of the numbers in the flow cytometry panels as suggested.

6. Was necrotic cell death estimated (AV-/PI+) for each treatment? If so the numbers should be provided more clearly in the flow figures. It is possible that the high taxane concentrations may have induced some necrotic cell death.

We have increased the size of the numbers in the flow cytometry panel Quadrant IV (necrosis) to show % of necrotic cell death as suggested. Necrotic cell death was ~5% in all control groups but increased to ~20% in the combination groups. The high taxane concentration is found to have little effect on inducing necrotic cell death. This is demonstrated in Figures 5 and 6.

7. Treatment with DTX or CBZ led to downregulation of the TRAIL receptors DR4 and DR5 in both PC3 and DU145 spheroids; however, 3D DU145 cultures were still sensitive to TRAIL + taxanes whereas PC3 cultures were more resistant, suggesting that downregulation of these receptors may not be related to the cell viability response. The authors argue that this difference may be related to the presence of more stem cells in the PC3 spheroids but the data is not presented. These contradictory results need more detailed discussion, and perhaps additional supportive data.

We have performed experiments and included data for the presence of more stem cells, This is presented in Figure 7 and discussed on page 21, lines 461-480.

8. To establish the role of the DR4 and DR5 receptors in influencing sensitivity to TRAIL + taxanes in PC3 and DU145 spheroids the authors should knockdown these receptors in 2D cells and examine culture sensitivity to this combinatorial treatment.

We have performed such experiments and included data to examine 2D cell sensitivity to the combinatorial treatment. Figures 1 and 2 and lines 285-313 discuss the influence. 

9. The cancer cells and the fibroblasts were stained with different Cell Tracker fluorescent dyes. How do the authors know that these dyes are specific for cancer cells or fibroblasts?

We stained cancer cell and fibroblasts separately in different 1.5mL Eppendorf tubes following the manufacturer’s instructions. Per the manufacturer: “CellTracker™ fluorescent dye has been designed to freely pass through cell membranes into cells, where it is transformed into cell-impermeant reaction products. CellTracker™ dye is retained in living cells through several generations. The dye is transferred to daughter cells but not adjacent cells in a population.” Based on this design, confocal images and multiple washes after staining, the dyes are specific to the cell population that is dyed with that color. 

10. It is not clear from the Methods if the PCa cells were mixed with the fibroblasts and then seeded as a mixture, or if the fibroblasts and the PCa cells were seeded sequentially. Regardless, it seems that the presence of fibroblasts does not influence cellular responses to TRAIL + taxanes. Discussion of this issue needs elaboration.

The PCa cells were mixed with the fibroblasts and then seeded as a mixture. We note in the Methods section page 7, lines 144-145. We state in lines 576-581 that the presence of fibroblasts did not enhance resistance or influence cellular responses to TRAIL + taxane but may serve more of an active role in influencing under high fluid shear stress unlike the low shear stress environment of a typical solid tumor. Other cell types such as immune cells or stem cells may play a more prominent role in response to TRAIL + taxane in these PCa cell lines. 

11. What are the comparison groups in Fig. 5, particularly for the bars with significant values? The statistical analysis needs to be clarified for this figure.

The significant values are being compared to the control groups. We have clarified this as suggested. 

12. There are several key statements in the Introduction and Discussion sections that need citations.

We have added references to the Introduction and Discussion session to address this.

---

## [Decision Letter · Decision Letter 1]

26 Jan 2021

Overcoming TRAIL-resistance by sensitizing prostate cancer 3D spheroids with taxanes

PONE-D-20-14145R1

Dear Dr. King,

We’re pleased to inform you that your manuscript has been judged scientifically suitable for publication and will be formally accepted for publication once it meets all outstanding technical requirements.

Kind regards,

Irina V. Lebedeva, Ph.D.

Academic Editor

PLOS ONE

Additional Editor Comments (optional):

Reviewers' comments:

Reviewer's Responses to Questions

**Comments to the Author**

1. If the authors have adequately addressed your comments raised in a previous round of review and you feel that this manuscript is now acceptable for publication, you may indicate that here to bypass the “Comments to the Author” section, enter your conflict of interest statement in the “Confidential to Editor” section, and submit your "Accept" recommendation.

Reviewer #1: All comments have been addressed

Reviewer #2: All comments have been addressed

Reviewer #3: All comments have been addressed

2. Is the manuscript technically sound, and do the data support the conclusions?

Reviewer #1: Yes

Reviewer #2: Yes

Reviewer #3: Yes

3. Has the statistical analysis been performed appropriately and rigorously? 

Reviewer #1: Yes

Reviewer #2: Yes

Reviewer #3: Yes

4. Have the authors made all data underlying the findings in their manuscript fully available?

Reviewer #1: Yes

Reviewer #2: No

Reviewer #3: Yes

5. Is the manuscript presented in an intelligible fashion and written in standard English?

Reviewer #1: Yes

Reviewer #2: Yes

Reviewer #3: Yes

6. Review Comments to the Author

Reviewer #1: (No Response)

Reviewer #2: No further comments necessary. The comments have been addressed.

No additional comments regarding ethics.

Reviewer #3: The authors have satisfactorily addressed all my earlier comments by including new data in the Figures and adding clarifications or explanations to the manuscript text.

7. PLOS authors have the option to publish the peer review history of their article (what does this mean?). If published, this will include your full peer review and any attached files.

Reviewer #1: No

Reviewer #2: No

Reviewer #3: No

---

## [Editor Report · Acceptance letter]

25 Feb 2021

PONE-D-20-14145R1 

Overcoming TRAIL-resistance by sensitizing prostate cancer 3D spheroids with taxanes 

Dear Dr. King:

I'm pleased to inform you that your manuscript has been deemed suitable for publication in PLOS ONE. Congratulations! Your manuscript is now with our production department. 

Kind regards, 

on behalf of

Dr. Irina V. Lebedeva 

Academic Editor

PLOS ONE